# Replication stress response in fission yeast differentially depends on maintaining proper levels of Srs2 helicase and Rrp1, Rrp2 DNA translocases

**Gabriela Baranowska**[1☯], **Dorota Misiorna**[1☯], **Wojciech Białek**[1], **Karol Kramarz**[2]*, **Dorota Dziadkowiec**[1]*

**1** Faculty of Biotechnology, University of Wrocław, Wrocław, Poland, **2** Faculty of Biological Sciences, Academic Excellence Hub—Research Centre for DNA Repair and Replication, University of Wrocław, Wrocław, Poland

☯ These authors contributed equally to this work.
* dorota.dziadkowiec@uwr.edu.pl (DD); karol.kramarz@uwr.edu.pl (KK)

**Data Availability Statement:** All relevant data are within the manuscript and its Supporting Information files.

## Abstract

Homologous recombination is a key process that governs the stability of eukaryotic genomes during DNA replication and repair. Multiple auxiliary factors regulate the choice of homologous recombination pathway in response to different types of replication stress. Using *Schizosaccharomyces pombe* we have previously suggested the role of DNA translocases Rrp1 and Rrp2, together with Srs2 helicase, in the common synthesis-dependent strand annealing sub-pathway of homologous recombination. Here we show that all three proteins are important for completion of replication after hydroxyurea exposure and provide data comparing the effect of overproduction of Srs2 with Rrp1 and Rrp2. We demonstrate that Srs2 localises to rDNA region and is required for proper replication of rDNA arrays. Upregulation of Srs2 protein levels leads to enhanced replication stress, chromosome instability and viability loss, as previously reported for Rrp1 and Rrp2. Interestingly, our data suggests that dysregulation of Srs2, Rrp1 and Rrp2 protein levels differentially affects checkpoint response: overproduction of Srs2 activates simultaneously DNA damage and replication stress response checkpoints, while cells overproducing Rrp1 mainly launch DNA damage checkpoint. On the other hand, upregulation of Rrp2 primarily leads to replication stress response checkpoint activation. Overall, we propose that Srs2, Rrp1 and Rrp2 have important and at least partially independent functions in the maintenance of distinct difficult to replicate regions of the genome.

## Introduction

Homologous recombination (HR) is crucial for maintaining genome stability due to its involvement in the repair of DNA double-strand breaks (DSBs) and recovery of arrested replication forks. In *S. pombe* the key Rad51 recombinase is assisted by auxiliary factors that

**Funding:** This work was supported by Preludium 13 (2017/25/N/NZ1/01974) grant from The National Science Centre to GB. KK was supported by the program „Excellence Initiative – Research University" for years 2020-2026 for University of Wrocław of the Ministry of Education and Science from Poland (grant number IDN.CBNDR 0320/2020/20). The funders had no role in designing the study, data collection and analysis, decision to publish, or preparation of the manuscript.

**Competing interests:** The authors declare no competing interests.

regulate its activity, and are evolutionary conserved not only in *S. cerevisiae* but also in humans [1]. Among them two complexes, Rad55-Rad57 and Swi5-Sfr1, were proposed to act in parallel to stabilize the Rad51 nucleofilament and stimulate Rad51-mediated strand exchange [2]. The choice of specific complex may determine the pathway and thus the final outcomes of HR, since it has been shown in *S. pombe* that double Holiday junction formation and channelling of the recombination intermediates into double strand break repair (DSBR) pathway is dependent on Rad55-Rad57, but does not require Swi5-Sfr1 [3]. We have previously described in *S. pombe* another complex, Rrp1-Rrp2 [4], and proposed it to function together with Swi5-Sfr1 and Srs2 helicase, but independently of Rad55-Rad57, in a synthesis dependent strand annealing pathway (SDSA) of HR [5]. Thus, according to our model, during recovery of replication forks arrested by hydroxyurea (HU) in the absence of Rad57-dependent branch of HR, Rad51 generated recombination products would be channelled into SDSA, away from the DSBR pathway requiring dissolution by Rqh1 (S1A Fig in S1 File). Since recombination intermediates requiring processing by Rqh1 would no longer be generated in the double *rqh1Δrad57Δ* mutant, its HU sensitivity should be alleviated, provided that the Rad57-independent pathway was functional. Indeed, we have observed that the presence of Rrp1 and Rrp2 was required for the full rescue of the *rqh1Δ* mutant's HU sensitivity and aberrant mitosis phenotype by deletion of *rad57+* [5].

Studies of the effect of upregulation of Rrp1 and Rrp2 protein levels allowed us to identify their novel biological functions in genome stability maintenance [6]. We have shown that Rrp1, and to a lesser degree Rrp2, are involved in the modulation of histone levels and their overproduction resulted in defects in centromere structure and function [7]. Based on the discovery that Rrp2 can protect SUMOylated Top2 from premature degradation [8], we have proposed that Rrp2 may protect telomeres against Top2-induced DNA damage, a function it does not share with Rrp1 [7]. Recently, we have demonstrated that Rrp1, but not Rrp2, can protect the cells from the toxicity of *rad51+* overexpression [9]. In addition, our *in vitro* and *in vivo* data showed that Rrp1 is a DNA-dependent translocase and ubiquitin ligase that can regulate Rad51 binding to dsDNA [9].

In summary, even though Rrp1 and Rrp2 have been established to act as a complex in replication stress response within a sub-pathway of HR, they can also have functions that are, at least partially, independent from each other.

Srs2 is a DNA-dependent helicase that belongs to the UvrD-like superfamily present in S. *cerevisiae* and *S. pombe*, with no direct homologue in human cells. Most of the information on Srs2 important roles in DNA replication, recombination and repair comes from work in *S. cerevisiae*, where it has been very extensively studied [10]. Several helicases, such as WRN, HELB, or PARI are regarded as potential functional human Srs2 orthologues, fulfilling some of the functions attributed to Srs2 in *S. cerevisiae*. Srs2 has been first described as "anti-recombinase" due to its ability to remove Rad51 from ssDNA [11,12]. Nevertheless, Srs2 has also been demonstrated to remove single strand DNA binding protein, RPA, from DNA and thus regulate checkpoint response [13,14]. Srs2 has also been shown to inhibit homologous recombination during post-replication repair in *S. cerevisiae* through its interaction with PCNA [15], although this activity seems not to be conserved in *S. pombe* [16]. Later however it has been proposed that Srs2 is rather involved in the determination of repair pathway choice and directs the processing of HR intermediates into the SDSA by several mechanisms unrelated to its interaction with Rad51: inhibiting D-loop extension through competition with delta polymerase, as well as direct D-loop unwinding [17,18]. This would fit with the proposed role for Srs2 in *S. pombe* based on genetic analyses [3,5], as described above and summarized in S1A Fig in S1 File.

Srs2 clearly is a multifunctional protein in *S. cerevisiae* and still more roles will probably be ascribed to it in the future. It is important to establish, which of these functions are conserved

in *S. pombe*, and which have been transferred to other proteins exhibiting helicase/translocase activities. This clarification would considerably help to elucidate the division of labour between functional orthologues of Srs2 in humans.

## Materials and methods

### Yeast strains, plasmids and general methods

Strains, plasmids and primers used in this study are listed in S1-S3 Tables, respectively. Media used for *S. pombe* growth were as described [19]. Yeast cells were grown in complete yeast extract plus supplements (YES) medium or glutamate supplemented Edinburgh minimal medium (EMM) at 28˚C. Thiamine at 5 µg/mL, geneticin (ICN Biomedicals) at 100 µg/mL and nurseotricin (Werner Bioagents) at 200 µg/mL were added where required. Multiple mutants were obtained by genetic crossing of relevant single mutants followed either by random spore analysis or by tetrad dissection. pREP81-FLAG plasmid carrying wild-type *srs2*+ was constructed using the Gibson Assembly® Cloning Kit (NEB E5510S) with primers listed in S3 Table in S1 File. After Gibson cloning, the construct was confirmed by sequencing and NdeI and SmaI digested insert was cloned into other required plasmids.

### Spot assays

Cells were grown to mid-log phase, then serially diluted by 10-fold and 2 µL aliquots were spotted onto relevant plates (YES or EMM) without drugs or plates containing camptothecin (CPT) or hydroxyurea (HU). Plates were incubated for 3–5 days at 28˚C and photographed. All assays were repeated at least twice.

### Survival assays

Logarithmic cultures of respective mutants for the survival of acute HU treatment assay were incubated for 4 hr with 12mM HU, diluted and plated on YES media. After incubation for 3–5 days at 28˚C the number of colonies formed was normalized by that for similarly treated wild-type strain. For Srs2 overproduction toxicity experiments transformants with *srs2*+ gene or empty vector control were grown for 48 hours in minimal medium with (repressed conditions) or without thiamine (overexpression) at 28˚C, serially diluted, plated onto YES medium and incubated for 3–5 days at 28˚C. The viable cells were counted and the percentage of survival for gene overexpression conditions was calculated against the repressed control.

### Yeast two-hybrid assay

Gal4-based Matchmaker Two-Hybrid System 3 (Clontech) was used according to the manufacturer's instructions. The indicated proteins were fused to the GAL4 activation domain (AD) in pGADT7 vector and the GAL4 DNA-binding domain (DBD) in pGBKT7, and expressed in the *S. cerevisiae* strain AH109. Transformants were selected on synthetic dextrose drop-out medium without Leu and Trp (SD DO-2), and then spotted on SD DO-2 as control and high stringency medium without Leu, Trp, His and Ade (SD DO-4). Plates were incubated for 3–5 days at 28˚C and photographed.

### Fluorescence microscopy

To determine the localization of overproduced EGFP-Srs2 and the influence of *srs2*+ overexpression on the accumulation of single stranded DNA, appropriate transformants in strains expressing tagged Rad11-EGFP or nucleolar histone chaperone Gar2-mCherry were grown for 24 h in EMM medium without thiamine. 1 mL of culture was harvested, washed with water

and subjected to fluorescent microscopy analysis. For examination of mitotic defects induced by *srs2*[+] overexpression, samples taken from respective transformant cultures grown for 48 hours in EMM medium without thiamine were fixed in 70% ethanol. After rehydration, cells were stained with 1 mg/mL 4',6-diamidino-2-phenylindole (DAPI) and 1 mg/mL p-phenylenediamine in 50% glycerol and examined by fluorescence microscopy. Images were captured under 100x magnification using Axio Imager A.2 (Carl Zeiss) and analysed with Axiovision rel. 4.8.

## Chromosome loss

Single white colonies from indicated transformants grown on EMM low Ade plates (adenine concentration 7.5 mg/L) with thiamine were inoculated into EMM medium without thiamine and incubated for 48 h at 28°C. Then cultures were diluted, plated on YES low Ade plates and incubated for 3–4 days at 28°C. The percentage of red to white colonies was then calculated as a readout for the loss of unessential mini-chromosome.

## Pulse field gel electrophoresis

Logarithmic yeast cultures (grown in rich YES medium) were diluted to $OD_{600}$ ~0.5, then exposed to 20 mM HU for 4 hours, subsequently washed with water and released to fresh YES medium. At indicated time points 20 mL of cell culture was collected, washed with cold 50 mM EDTA pH 8 and digested with litycase (Sigma, L4025) in CSE buffer (20 mM citrate/phosphate pH 5.6, 1.2 M sorbitol, 40 mM EDTA pH 8). Spheroplasts were embedded into 1% UltraPure[TM] Agarose (Invitrogen, 16500) and put into 4 agarose plugs per each time point. Obtained plugs were incubated in Lysis Buffer 1 (50 mM Tris-HCl pH 7.5, 250 mM EDTA pH 8, 1% SDS) for 90 minutes at 55°C and then digested in Lysis Buffer 2 (1% N-lauryl sarcosine, 0.5 M EDTA pH 9.5, 0.5 mg/mL proteinase K) o/n at 55°C. Lysis Buffer 2 was changed the next morning for a fresh one, and digestion was continued o/n at 55°C. Plugs were then stored at 4°C. Pulse field gel electrophoresis was carried out on Biorad CHEF-DR-III system for 48 hours at 2.0 V/cm, with an angle 120°, at 14°C. Single switch time was set at 1800 s, pump speed 70. Electrophoresis was carried out in 1x TAE buffer. Chromosomes were visualized at Biorad Chemidoc MP after gel staining in ethidium bromide (10 μg/mL) for 30 min and washing for 30 min in 1x TAE.

## Flow cytometry

The analysis of DNA content by flow cytometry was performed as follows [20]: cells, fixed in 70% ethanol and washed with 50 mM sodium citrate, next were digested by RNAse A (Sigma, R5503) for 2 h, and stained with 1 μM Sytox Green (Invitrogen, S7020). Samples were subjected to flow cytometry using Guava easyCyte (Millipore).

## Two-dimensional gel electrophoresis analysis of rDNA locus

Replication intermediates analysis was performed as described earlier [21] with following modifications. Logarythmic cultures (2.5 x 10$^9$ cells per sample) were treated with 0.1% sodium azide and then 50 mL of 0.5 M EDTA pH 8 was added to each culture. Cells were exposed to trimethyl psoralen crosslink (0.01 mg/mL TMP, Sigma, T6137) for 5 min in the dark and subsequently irradiated with UV-A (365 nm), 10 mW/cm$^2$ for 8 min. Next, samples were digested with lysing enzymes (Sigma, L1412, 0.625 mg/mL) and zymolyase 100T (Amsbio, 120493–1, 0.5 mg/mL). Spheroplasts were embedded into low melting agarose (NuSieve[TM] GRG[TM] Agarose, 50081, Lonza, final concentration 1%). Plugs were digested overnight

at 55˚C with 1mg/ml proteinase K. Digested plugs were washed 3 times in 50x TE buffer (10 mM Tris-HCl pH 7.5, 50 mM EDTA pH 8) and stored at 4˚C. Plugs were washed in 1x TE (10 mM Tris-HCl pH 7.5, 1 mM EDTA pH 8) and digested with 60 units per plug of *BamHI* (NEB, R0136T). After overnight digestion, plugs were melted and treated with RNAse (Roche, 11119915001) and beta-agarase (NEB, M0392L). Replication intermediates were purified on BND-cellulose (Sigma, B6385) in columns (Biorad, 731–1550). Replication intermediates were eluted in the presence of 1M NaCl, 1.8% caffeine (Sigma, C-8960), subsequently precipitated with glycogen (Roche, 1090139001) and migrated in 0.4% ultrapure agarose gel in 1x TBE for first dimension. The slices of DNA relevant for *BamHI* 3.1 kB fragment of rDNA were cast for second dimension in 1% ultrapure agarose gel in presence of ethidium bromide. The DNA was capillary transferred onto neutral nylon membrane (GE Healthcare RPN203N) in 10x SSC. Membranes were incubated with 1354 bp rDNA probes (amplified with primers listed in S3 Table in S1 File) and chemiluminescent kit AlkPhos Direct Labeling Module (Cytiva, RPN3680) with CDP-star detection reagent (RPN3682). Signal of replication intermediates was collected in Chemidoc MP (Biorad).

## Chromatin immunoprecipitation

EGFP-Srs2 ChIP was performed as described in [20] with the following modifications. 100 mL of exponential culture (OD600 ~1) was crosslinked with 10 mM DMA dimethyl adipimidate, Thermo scientific, 20660) and subsequently 1% formaldehyde (Sigma, F-8775). Cells were frozen in liquid nitrogen and extracted in lysis buffer 50 mM HEPES pH 7.5, 1% Triton X100, 0.1% Nadeoxycholate, 1 mM EDTA with 1 mM PMSF and Complete EDTA-free protease inhibitor cocktail tablets (Roche, 1873580). Chromatin sonication was done in a Diagenode Bioruptor Pico using Easy mode 10 cycles of 30 s ON and 30 s OFF. Immunoprecipitation over night was performed as follows: 300 μL was incubated with anti-GFP antibody (Invitrogen, A11122) at 1:150 concentration, and 5 μL was preserved as INPUT fraction. Then Protein G Dynabeads (Invitrogen, 10003D) were added for 1 h and immunoprecipitated complexes were decrosslinked for 2 h at 65˚ C. The DNA associated with Srs2 was purified with a Qiaquick PCR purification kit (QIAGEN, 28104) and eluted in 300 μL of water. qPCR (SsoAdvanced Universal SYBRⓇ Green Supermix, #1725274, primers listed in S3 Table in S1 File) was performed to determine the Ct using BIORAD CFX Maestro v1.1. Srs2 enrichment was calculated as % of INPUT by subtraction values obtained for the intergenic locus on chromosome II.

## Whole protein extract analysis

The trichloroacetic acid (TCA) method was used to obtain protein extracts. Mid-logarithmic cultures ($\sim 10^8$) of indicated strains were harvested after 24 hours of induction of the *nmt* promoter by removal of thiamine from media and lysed with lysis buffer (2 M NaOH, 7% β-mercaptoethanol). Total protein was precipitated with 50% TCA. Pellet was resuspended in 1 M Tris at pH 8 and 4x Laemmli buffer was added (250 mM Tris-HCl, pH 6.8, 8% SDS, 20% glycerol, 0.02% Bromophenol blue, 7% β-mercaptoethanol). Samples were analysed by SDS-PAGE and Western blotting using anti-FLAG (Sigma-Aldrich, F1804), anti-H2A (Active Motif, 39235), anti-γ-H2A (Abcam, ab15083), anti-HA (Roche, 11538816001) antibodies. Ponceau S staining (Sigma-Aldrich) of blotted membranes was used as loading controls. Image Lab (Western blots) or ImageJ software (Ponceau S staining) were used for protein quantification. Relative intensity was calculated by dividing sample intensities by the mean of control intensities obtained for each blot (details for each experiment are provided in figure captions). For

each experiment data from at least two different transformants from two independent protein isolations were analysed.

## Checkpoint activation

Cells were grown for 48 hours in EMM minimal medium without thiamine (over-expression conditions) at 28°C. 1 mL samples were taken from each culture and subjected to microscopy analysis in order to determine cell length. Images were captured under 100x magnification using Axio Imager A.2 (Carl Zeiss) and analysed with Axiovision rel. 4.8. From the remaining cultures whole cell protein extracts were isolated and subjected to Western blot analysis described above in order to determine levels of H2A and Chk1 phosphorylation.

## Statistical data analysis

Student's t test was used to calculate the P-values (**** $p \leq 0.0001$, *** $p \leq 0.001$, ** $0.001 < p \leq 0.01$, * $0.01 < p \leq 0.05$).

## Results

### Srs2 helicase may have a function in a similar HR repair pathway to Rrp1 and Rrp2

We have previously demonstrated that Rrp1 and Rrp2 are involved in replication stress response and may act together with Srs2 helicase within the Swi5-Sfr1 branch of the synthesis-dependent strand annealing homologous recombination repair pathway [5]. Additionally we found that the presence of Rrp1 and Rrp2 was required for the full rescue of the *rqh1Δ* mutant's HU sensitivity and aberrant mitosis phenotype by deletion of *rad57+* [5]. Here we show that the deletion of *srs2+* also increases the double *rad57Δrqh1Δ* mutant's HU sensitivity, to the same level as the deletion of *rrp1+* or *rrp2+* (S1B Fig in S1 File). *rqh1Δ* mutant is also sensitive to acute HU treatment as seen by the drop in its survival relative to wild-type strain after 4 h incubation in 12mM HU (S1C Fig in S1 File). This phenotype is partially rescued by deletion of *rad57+* in a manner dependent on the presence of Srs2. Furthermore, deletion of *rad57+* in in *rqh1Δ* mutant results in a decrease of the number of cells with nuclear defects, especially with cut phenotype (S1D Fig in S1 File purple arrowhead), but not lagging or mis-segregated chromosomes (S1D Fig in S1 File orange and green arrowheads, respectively). The incidence of these aberrant mitotic phenotypes is increased in *rad57Δrqh1Δsrs2Δ* mutant compared to *rad57Δrqh1Δ* (S1E Fig in S1 File), similar to what has been shown earlier for *rad57Δrqh1Δrrp1Δ* and *rad57Δrqh1Δrrp2Δ* [5]. This lends support to our published model placing Rrp1, Rrp2 and Srs2 in a common pathway.

We have previously shown that while cells devoid of *srs2+* were slightly sensitive to hydroxyurea (HU) the single mutants *rrp1Δ* and *rrp2Δ* were not [5]. However, when the ability to resume replication upon transient treatment of studied mutants with 20 mM HU was examined by pulse field gel electrophoresis we found that not only the presence of Srs2 but also of Rrp1 and Rrp2 was required for proper replication completion (Fig 1A). Wild-type (WT) cells completely resolved replication intermediates 90 minutes after incubation in drug-free medium, indicated by the doubling of the intensity of chromosomes compared to initial asynchronous culture. In *srs2Δ* cells and (somewhat surprisingly given their lack of HU sensitivity) in both *rrp1Δ* and *rrp2Δ* mutants the increase in chromosomes intensities at 90 min time point was far less pronounced (Fig 1A). The double mutants simultaneously devoid of *srs2+* and *rrp1+* or *rrp2+* did not show any further delay in the entry of chromosomes into the gel (Fig 1B) suggesting that Srs2, Rrp1 and Rrp2 may have overlapping roles in the resolution of

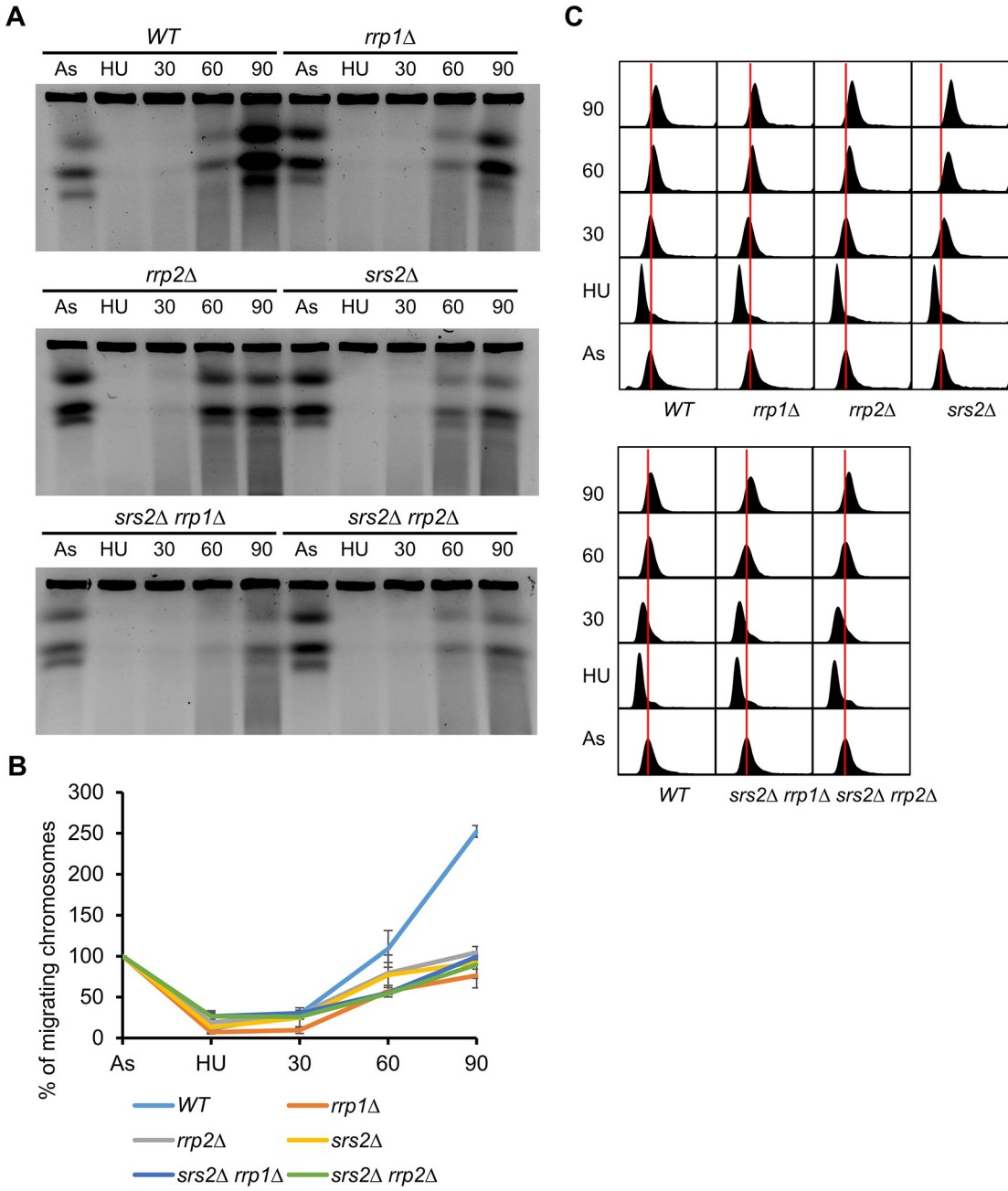

**Fig 1. Srs2 helicase and Rrp1 and Rrp2 DNA translocases are required for resolution of replication intermediates.** (A) Chromosome analysis by pulse field gel electrophoresis (PFGE) in respective mutants at indicated time points: As–asynchronous, logarithmic cells exposed to 20 mM HU by 4 hours (HU) and after drug removal released into fresh medium at 30°C to verify ability to resolve replication intermediates at indicated time points). (B) Quantification of % of chromosomes migrating in gel from studied time points. Values are means of two independent biological experiments. Error bars are deviation from the mean. (C) Flow cytometry analysis of DNA content in cultures of respective mutants used for PFGE.

replication intermediates after HU block. Wild-type cells and all studied mutants reach G2 stage of cell cycle after 90 minutes recovery from HU block as seen in FACS analysis of samples used for the PFGE experiment (Fig 1C), suggesting that the observed persistence of replication intermediates does not stem from substantial differences in the length of S-phase.

Previously we have found that even though *rrp1Δ* and *rrp2Δ* single mutants displayed very subtle phenotypes, over-expression of *rrp1+* and *rrp2+* genes allowed us to identify novel biological functions for both proteins, at least partially independent from each other [7]. We thus decided to perform similar analysis for Srs2 protein.

## Upregulation of Srs2 protein levels leads to growth defects independent of Rad51, Rrp1 and Rrp2

It has been demonstrated in *S. cerevisiae* that an increase in Srs2 copy number is toxic in certain mutant contexts [22] and we have shown that overproduction of Rrp1 or Rrp2 causes viability loss in wild-type cells grown under unperturbed conditions [7]. In order to investigate the effect of *srs2*$^+$ over-expression on cell growth, we have cloned the *srs2+* gene under the *nmt* promoter into several pREP expression plasmids (S2A Fig in S1 File). The Srs2 protein was produced even when the *nmt* promoter was repressed by addition of thiamine (S2B Fig in S1 File). Overproduced EGFP-Srs2 accumulated in the nucleus, as expected, and is mostly enriched in one patch, as is characteristic for rDNA repeats (S2C Fig in S1 File, blue arrowhead). In some cells, distinct EGFP-Srs2 foci could also be observed (S2C Fig in S1 File, white arrowhead). Finally, obtained constructs were able to complement the CPT and HU sensitivity of *srs2Δ* mutant (S2D Fig in S1 File).

The complementation of genotoxin sensitivity was greater when the *nmt* promoter was repressed (S2D Fig in S1 File leu+thi), resulting in much lower amount of Srs2 produced (S2B Fig in S1 File, compare + thi and -thi lanes) suggesting that upregulation of Srs2 protein levels might also be toxic in *S. pombe*. Indeed, *srs2+* over-expression from the medium strength *nmt41* promoter resulted in the growth defect and viability loss, although not as pronounced as those for *rrp2+*, but comparable to those seen for *rrp1+* (Fig 2A and 2B) [7]. The levels of overproduced Srs2 protein was somewhat higher than Rrp1 and Rrp2 (Fig 2C) suggesting that toxicity observed is not directly proportional to proteins levels in the cell. Toxicity of Srs2 overproduction was not dependent on the presence of Rrp1 and seemed only slightly more pronounced in cells lacking Rrp2 and Rad51 recombinase (Fig 2D and 2E). In *S. cerevisiae* it has also been shown that toxicity of Srs2 overproduction was not mediated by Rad51 [22]. Similarly, growth defect induced by *rrp1+* or *rrp2+* over-expression in the *srs2Δ* mutant was similar to that seen in wild-type (Fig 2F). Our yeast-two-hybrid analysis suggests that Srs2 does not form a direct complex with Rrp1 or Rrp2 (Fig 2G). It is important to note, however, that such interaction may be mediated through other proteins, and needs to be investigated further. Nevertheless, the above data indicate that the effects of overexpression of these three genes may be, at least in part, independent of each other.

## Srs2 is important for proper replication of rDNA region

Our initial data suggested that overproduced Srs2 may be localised in the same region as rDNA repeats (S2C Fig in S1 File). In order to confirm this we overproduced EGFP-Srs2 in a strain expressing Gar2-mCherry, a nucleolar histone chaperone and observed co-localisation of both proteins in more than 80% of cells (Fig 3A blue arrowhead), consistent with Srs2 being previously shown to be important for rDNA stability [23]. We also observed that EGFP-Srs2 forms distinct foci, mainly at the border and outside of the nucleolus, sometimes quite numerous and very bright (Fig 3A, white arrowhead), suggestive of Srs2 accumulation also in other regions of the genome, possibly experiencing replication stress. Finally, we have confirmed by chromatin immunoprecipitation of EGFP-Srs2 followed by qPCR analysis that when overproduced Srs2 accumulates at DNA within the rDNA region (Fig 3B).

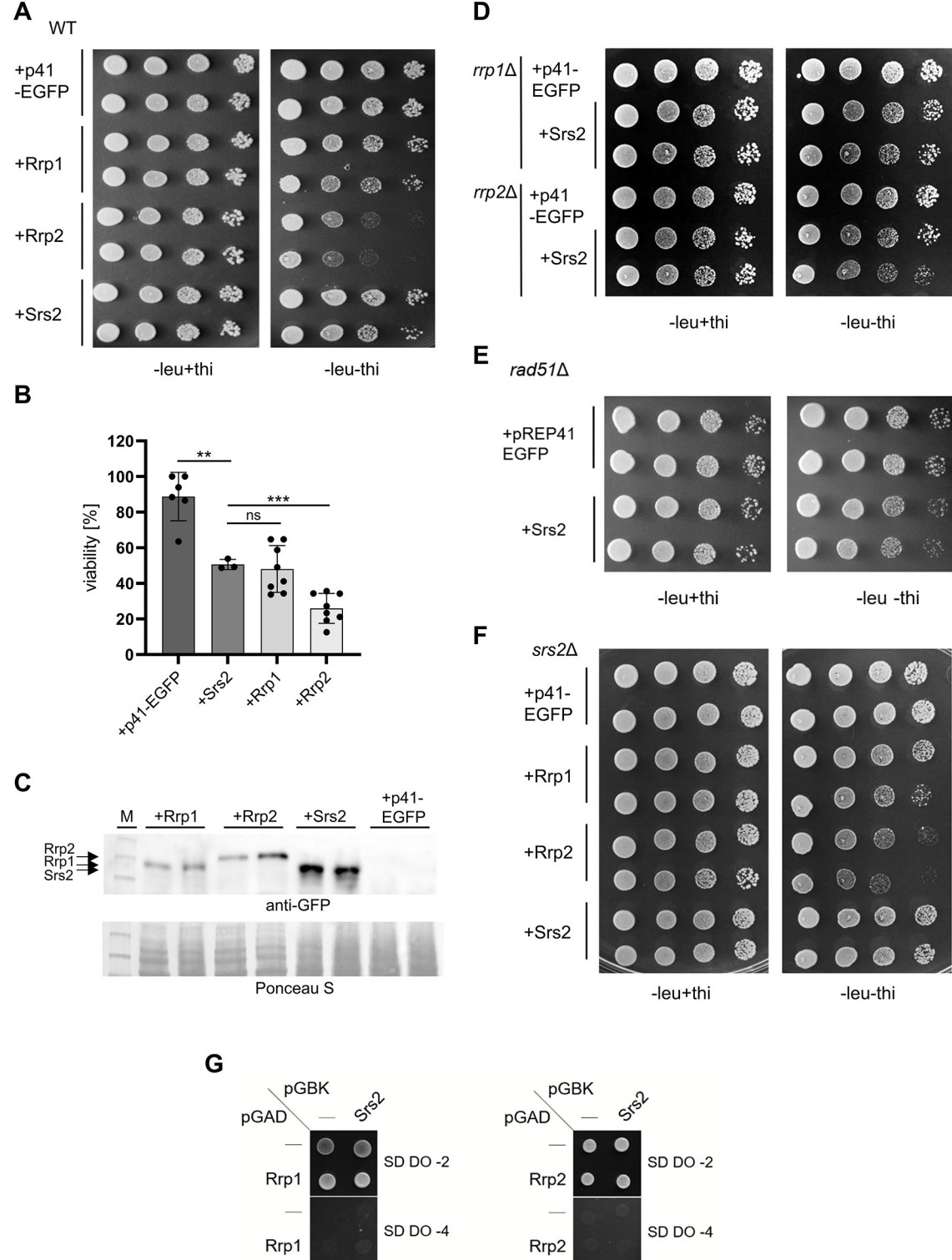

**Fig 2. Upregulation of Srs2 protein levels leads to growth defect.** Induction of *srs2+* expression leads to growth defect (A) and viability loss (B) comparable to that seen for *rrp1+* overexpression. Cells of respective transformants were appropriately diluted and spotted on EMM plates where *srs2+* expression was induced (-leu) or repressed (-leu + thi), incubated for 6 days and photographed. For viability the ratio of surviving cells of indicated transformants grown under inducing conditions to those grown without induction was determined. The experiment was repeated for at least three independent transformants. Error bars represent the

standard deviation about the mean values. Student's t-test was performed to calculate P-values (*** P ≤ 0.001, ** 0.001 < P ≤ 0.01, ns 0.05≤P). (C) All proteins are expressed, albeit at slightly different levels, as confirmed by Western blot with anti-GFP antibodies of proteins isolated from cultures incubated for 24 h without thiamine (expression inducing conditions). Deletion of *rrp1+* or *rrp2+* (D) nor *rad51+* (E) did not have a major effect on growth defect induced by overproduction of Srs2. (F) Overexpression of *rrp1+*, *rrp2+* and *srs2+* was not affected by the lack of *srs2+* gene. (G) Srs2 does not interact directly with Rrp1 or Rrp2 as seen in yeast-two hybrid system. Transformants were selected on synthetic dextrose drop-out medium without Leu and Trp (SD DO-2), then plated on high stringency medium without Leu, Trp, His and Ade (SD DO-4).

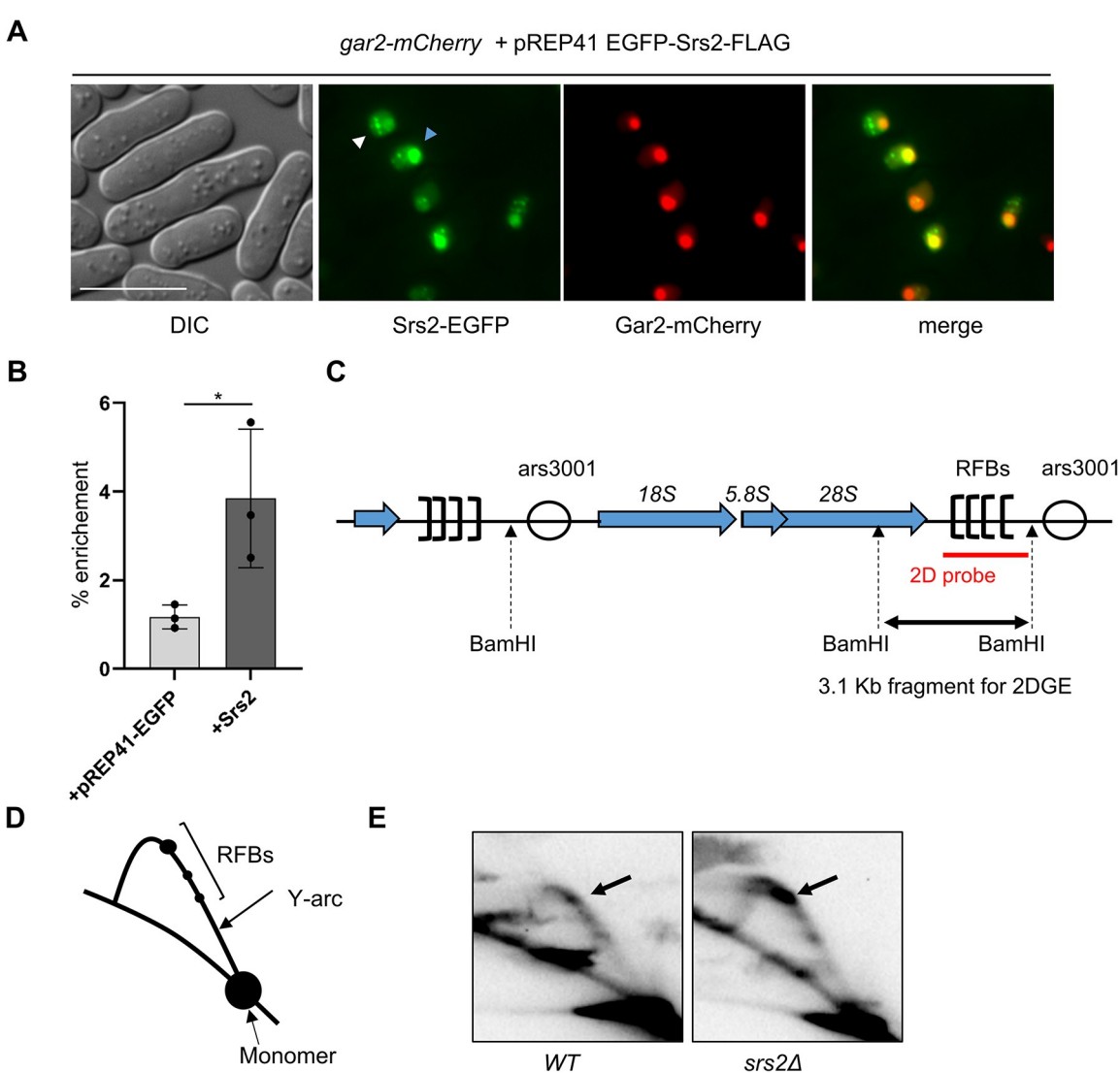

**Fig 3. Srs2 protein is important for rDNA replication.** (A) Overproduced EGFP-Srs2 co-localizes with Gar2-mCherry in the nucleolus (blue arrowhead) and also forms spontaneous foci (white arrowhead). Cells from induced cultures in a strain expressing nucleolar Gar2-mCherry protein were analysed by fluorescence microscopy. Scale bar indicates 10 μM (B) Chromatin immunoprecipitation of Srs2 at rDNA locus in strains expressing Srs2-EGFP from pREP41-EGFP compared to empty vector control. Error bars represent the standard deviation about the mean values from three experiments. Student's t-test was performed to calculate P-values (* 0.01 < *p* ≤ 0.05). (C) A diagram depicting rDNA locus and position of replication fork barriers (RFBs) within *BamHI* digestion sites utilized for two-dimensional gel electrophoresis (2DGE) and the probe used for Southern blot analysis. (D) A diagram of replication profile (Y-arc) obtained by 2DGE of rDNA locus. RFBs–replication fork blocks. Monomer– 3.1 Kb *BamHI*-released fragment from genomic DNA. (E) Southern blot analysis of rDNA replication in wild-type (WT) cells and srs2Δ mutant (black arrows indicate major replication arrest site).

In order to directly examine, if Srs2 has a role in the progression of replication through replication fork barriers (RFBs) present in rDNA locus we have employed two-dimensional gel electrophoresis (2DGE) approach using an rDNA fragment released by *BamHI* digestion from genomic DNA extracted from wild-type and *srs2Δ* cells (Fig 3C). 2DGE technique allows the visualization of replication profiles of selected locus (Y-arc, Fig 3D). Pausing of replication forks encountering obstacles, such as RFBs in rDNA arrays, is visible on Y-arc as black spots (Fig 3D). In 2DGE analysis of wild-type cells we can observe accumulation of arrested replication forks at endogenous RFBs in rDNA, however in cells devoid of *srs2+*, the stalling of replication forks at rRFBs is much more pronounced, visible as larger spots on Y-arc (Fig 3E, main stall site is marked with black arrow in both backgrounds). Thus, we have shown *in vivo* that Srs2 binds to rDNA and is important for unperturbed replication of the rDNA locus.

## Upregulation of Srs2 protein levels leads to replication stress, chromosome instability and viability loss

We have previously observed that *rrp1+* and *rrp2+* over-expressing cells accumulated exceptionally bright Rad11 (RPA) foci which may indicate excessive accumulation of single-stranded DNA (ssDNA), as well as fragmented DNA and Rad11 coated bridges [7], which together suggest that these cells suffer chromosome segregation defects. Upregulation of Srs2 also led to the appearance of such foci, but only in about ~3% of cells (Fig 4A and 4B, orange box). The increase in the total Rad11 foci number was also very modest upon Srs2 overexpression, compared to spontaneous foci visible in empty vector control (Fig 4B). This was accompanied by the accumulation of mitotic aberrations, detected by microscopic examination of DAPI-stained cells, such as mis-segregated and cut chromosomes (Fig 4C green and purple arrowheads, respectively) to a level that was comparable to that reported previously for cells overproducing Rrp1, but much lower than that observed for Rrp2, (Fig 4D) [7].

Chromosome instability in cells overproducing Srs2 was assessed using a strain with a non-essential Ch16 mini-chromosome carrying the ade6-216 allele trans-complementing the endogenous ade6-210 allele of the host cell. Cells that lost this mini-chromosome accumulate red intermediate due to the inability to complete adenine synthesis (Fig 4E). By scoring the number of red colonies on the medium with a limiting concentration of adenine we observed an increase in the loss of Ch16 mini-chromosome in *srs2+* over-expressing cells (Fig 4F), but two fold lower than reported earlier for Rrp1 and Rrp2 overproduction [7].

Overall, the effects of the upregulation of Srs2 are more similar to those for Rrp1 but less severe, especially than those caused by *rrp2+* over-expression. Together with the fact that over-produced Srs2 seems to localize predominantly in the nucleolus (Fig 3A), while Rrp1 and Rrp2 are mainly present at centromeres and telomeres, respectively [7], the above data raise the possibility that upregulation of Srs2 levels may affect the cells in a way distinct from these translocases.

## Differential checkpoint activation by upregulation of Srs2, Rrp1 and Rrp2

Accumulation of ssDNA and lagging chromosomes that break during mitosis is expected to result in checkpoint activation [24]. We thus reasoned that dysregulation of Srs2, Rrp1, and Rrp2 protein levels should result in DNA damage and/or replication checkpoint pathway activation. This can be readily observed in *S. pombe* as an increase in cell length and indeed cells over-expressing *srs2+* as well as *rrp1+* or *rrp2+* were elongated (Fig 5A). Rad3, a protein related to mammalian checkpoint kinase ATR, is a key *S. pombe* kinase activated upon DNA damage or replication arrest, and phosphorylates its effector kinases, Chk1 and Cds1, responsible for activation of DNA damage and replication checkpoints, respectively [25]. Rad3 is also

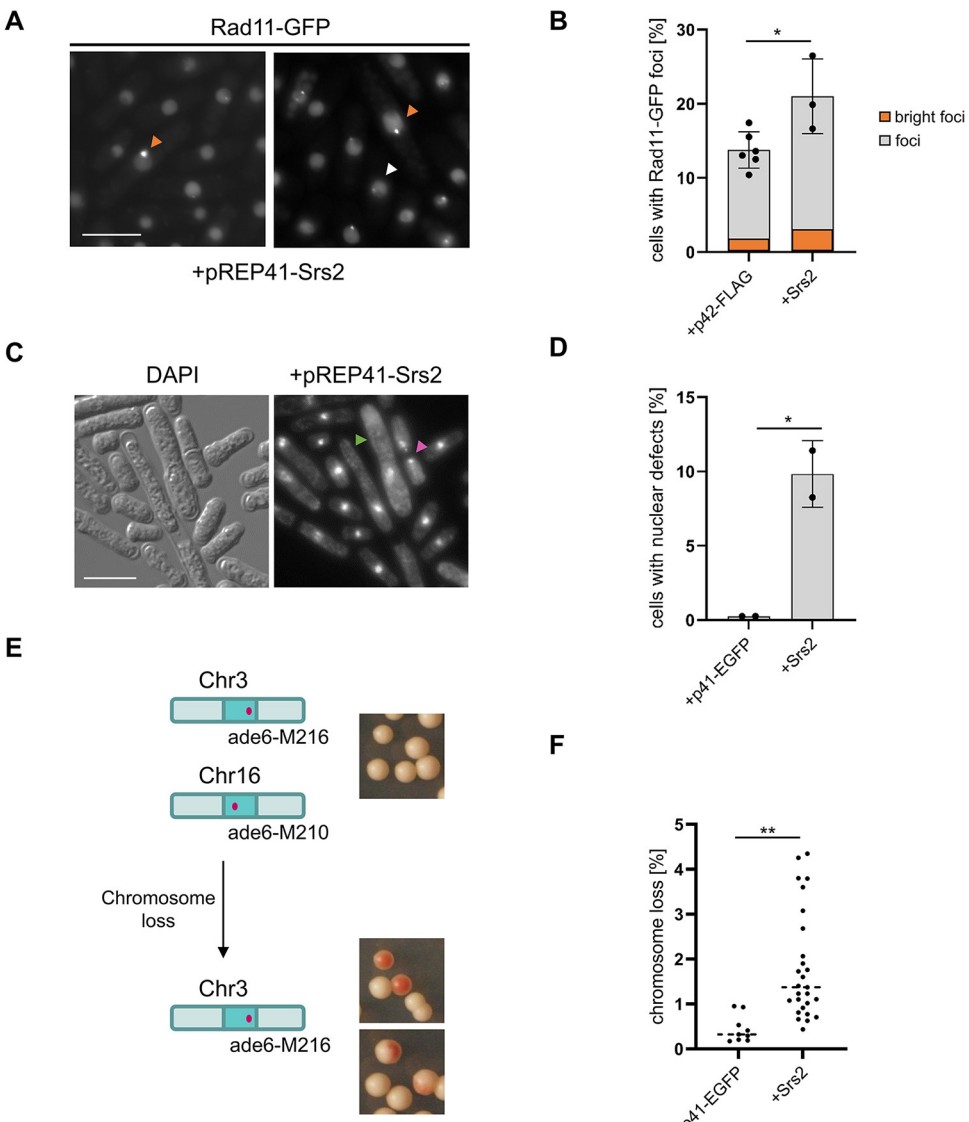

**Fig 4. Upregulation of Srs2 protein level leads to chromosome instability.** (A) Bright Rad11 foci (examples marked with orange arrowheads) accumulate in *srs2+* overexpressing cells. The level of regular Rad11 (white arrowheads) also increase (B). The experiment was repeated for at least three independent transformants. Error bars represent the standard deviation about the mean values. Student's t-test was performed to calculate P-values (* $0.01 < P \leq 0.05$). (C) Mitotic aberrations observed by DAPI staining of the nuclei of wild-type transformants grown for 48 h in EMM under expression- inducing conditions (-leu) accumulate in cells over-expressing *srs2+*. Cut and mis-segregated chromosomes are indicated by purple and green arrowheads, respectively. Scale bar represents 10 μM. (D) Two independent transformants for vector and *srs2+* were analysed and the total number of cells counted was above 900. Error bars represent the deviation from the mean values. Student's t-test was performed to calculate P-values. (* $0.01 < P \leq 0.05$). (E) Induction of *srs2+* expression leads to the loss of the nonessential Ch16 minichromosome carrying the ade6-216 allele, resulting in red colony formation on medium with limiting adenine concentration. (F) A total of 9 independent transformant colonies for vector and 27 for Srs2 from two independent transformations were analysed. Error bars represent the deviation from the mean values. Student's t-test was performed to calculate P-values. (** $0.001 < P \leq 0.01$).

responsible for phosphorylation of histone H2A in fission yeast in response to DNA damage [26] but also when replication fork stall at various barriers, repetitive DNA and heterochromatin in the centromeres and telomeres, as well as ribosomal DNA (rDNA) [27]. We found that

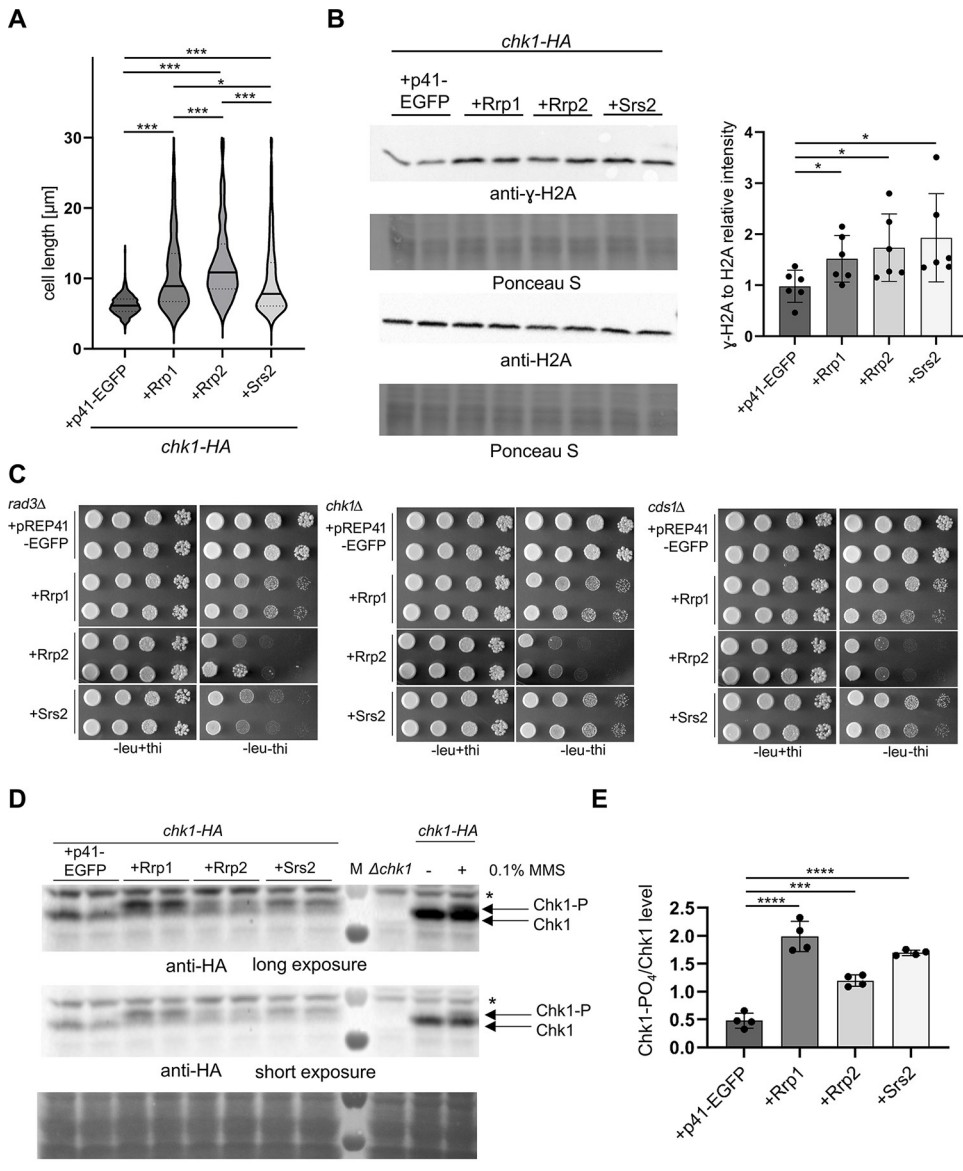

**Fig 5. Checkpoint response to upregulation of Srs2, Rrp1 and Rrp2.** (A) Over-expression of *srs2+*, *rrp1+*, *rrp2*
+ leads to an increase in cell length. Respective transformants in the Chk1-HA strain were grown for 48h in EMM
under expression-inducing conditions (-thi), observed under the microscope and analysed with Axiovision rel. 4.8.
Data from two independent transformations were analysed and the total number of cells counted was above 375.The
centre line represents the median; the dotted lines represent upper and lower quartiles. Student's t-test was performed
to calculate P-values (*** P ≤ 0.001, * 0.01 < P ≤ 0.05). (B) ɣH2A phosphorylation increases upon overproduction of
all studied proteins. Total protein extracts were collected after 24h growth in EMM-leu to induce the *nmt* promoter
and analysed by Western blot using anti- ɣH2A and anti-H2A antibodies. Data were quantified and shown as the
intensity of ɣH2A signal versus H2A signal. Reads were normalised by the mean value obtained for vector control
samples. Western blots from at least two separate protein isolations from two different transformants were examined.
Error bars represent the standard deviation about the mean values. Student's t-test was performed to calculate P-values
(* 0.01 < P ≤ 0.05). (C) Deletion of *rad3+* only aggravated the growth defect induced by *srs2+* over-expression. Cells
of respective transformants were appropriately diluted and spotted on EMM plates where expression of respective
genes was induced (-leu) or repressed (-leu + thi), incubated for 6 days and photographed. (D) Over-expression *srs2+*,
*rrp1+*, *rrp2+* leads to the increase in Chk1 phosphorylation. Total protein extracts were collected after 24h induction of
*nmt* promoter and analysed by Western blot. Chk1 was detected with anti-HA antibody and its phosphorylated form is
visible as slower migrating band (Chk1-PO$_4$, unspecific band is marked as *). *chk1Δ* and Chk1-HA strains as well as
Chk1-HA strain treated for 1 hour with 0.1% MMS were used as controls. (E) Data were quantified and shown as ratio
of Chk1-PO$_4$ signal over Chk1 signal. The experiment was repeated twice for two independent transformants. Error
bars represent the standard deviation about the mean values. Student's t-test was performed to calculate P-values (****
P ≤ 0.0001, *** P ≤ 0.001).

in cells over-expressing *srs2+* as well as *rrp1+* or *rrp2+* the level of γH2A was increased (Fig 5B), thus confirming the activation of the checkpoint response.

Deletion of *rad3+* aggravated the growth defect induced by *srs2+* over-expression, however deletion of *cds1+* or *chk1+* had no visible effect (Fig 5C), indicating that DNA damage and replication checkpoint pathways may have redundant roles in augmenting the survival of cells overproducing Srs2. This is different from what has been observed in *S. cerevisiae* where toxicity of *SRS2* over-expression significantly increased when DNA replication, but not DNA damage, checkpoint was compromised [22]. Growth defect induced by Rrp1 and Rrp2 overproduction in all 3 checkpoint mutants was comparable to that in wild-type cells (Fig 5C). This indicates that the checkpoint response might be affected in different ways upon upregulation of Srs2, Rrp1 and Rrp2.

Indeed, we observed that *srs2+*, *rrp1+* or *rrp2+* over-expression resulted in Chk1 phosphorylation, albeit to different levels (Fig 5D) that did not correlate with severity of growth defect induced by increased levels of these proteins (seen in Fig 2A), suggesting different contributions of DNA damage and replication stress response to checkpoint activation. The toxicity of *rrp2+* overexpression was the most severe, yet resulted in the mildest activation of Chk1. In contrast, phosphorylation of Chk1 was most apparent in *rrp1+* over-expressing cells, with Srs2 overproduction having the intermediate effect (Fig 5E).

We thus decided to assess the relative contribution of Chk1 and Cds1 pathways by measuring the cell length of respective checkpoint mutants, overproducing Srs2, Rrp1 or Rrp2 proteins as compared to wild-type. As expected, we found that in the *rad3Δ* mutant overexpression of all three proteins failed to induce checkpoint activation measured as an increase in cell length relative to empty vector control. Cell length increased relative to the *rad3Δ* mutant for all three proteins when overproduced in *chk1Δ* and *cds1Δ* mutants (Fig 6A–6C), demonstrating that both DNA damage and replication stress response pathways were activated. However, the relative contributions of these pathways in cells overproducing Srs2, Rrp1 or Rrp2, varied. For Srs2 the increase in cell length was only slightly greater in *cds1Δ*, than in *chk1Δ* mutant, indicating that checkpoint activation was similarly dependent on both pathways (Fig 6A). The increase in cell length in *cds1Δ* as compared to the *chk1Δ* mutant was much more apparent upon *rrp1+* over-expression, indicating that checkpoint response in these cells was more dependent on DNA damage effector kinase Chk1 (Fig 6B). Conversely, for *rrp2+* over-expression the increase in cell length was greater in *chk1Δ* mutant (Fig 6C), suggesting that checkpoint response in these cells was more dependent on the replication stress effector kinase Cds1. These results are consistent with levels of Chk1 phosphorylation observed in wild-type cells: intermediate for Srs2, highest for Rrp1 and lowest for Rrp2 overproduction (Fig 5D and 5E).

Srs2 has been shown to work in a pathway redundant to Rad3 and Mrc1, possibly independent from checkpoint signalling functions of these proteins but requiring Srs2 ATPase activity [23]. Consistently, we observed that the double *chk1Δsrs2Δ* mutant was more sensitive to HU and CPT than single *chk1Δ* mutant (Fig 6D), while no additive effect on sensitivity for double *cds1Δsrs2Δ* mutant was seen (Fig 6E). This suggests that Srs2 has an important role in preventing DNA damage during replication and this becomes especially apparent when DNA damage checkpoint pathway cannot be activated.

## Discussion

Rrp1 and Rrp2 have been shown to be required for the full rescue of the *rqh1Δ* mutant's HU sensitivity and aberrant mitosis phenotype by deletion of *rad57+* and proposed to act together with Swi5 and Srs2 in a synthesis-dependent strand annealing HR repair pathway

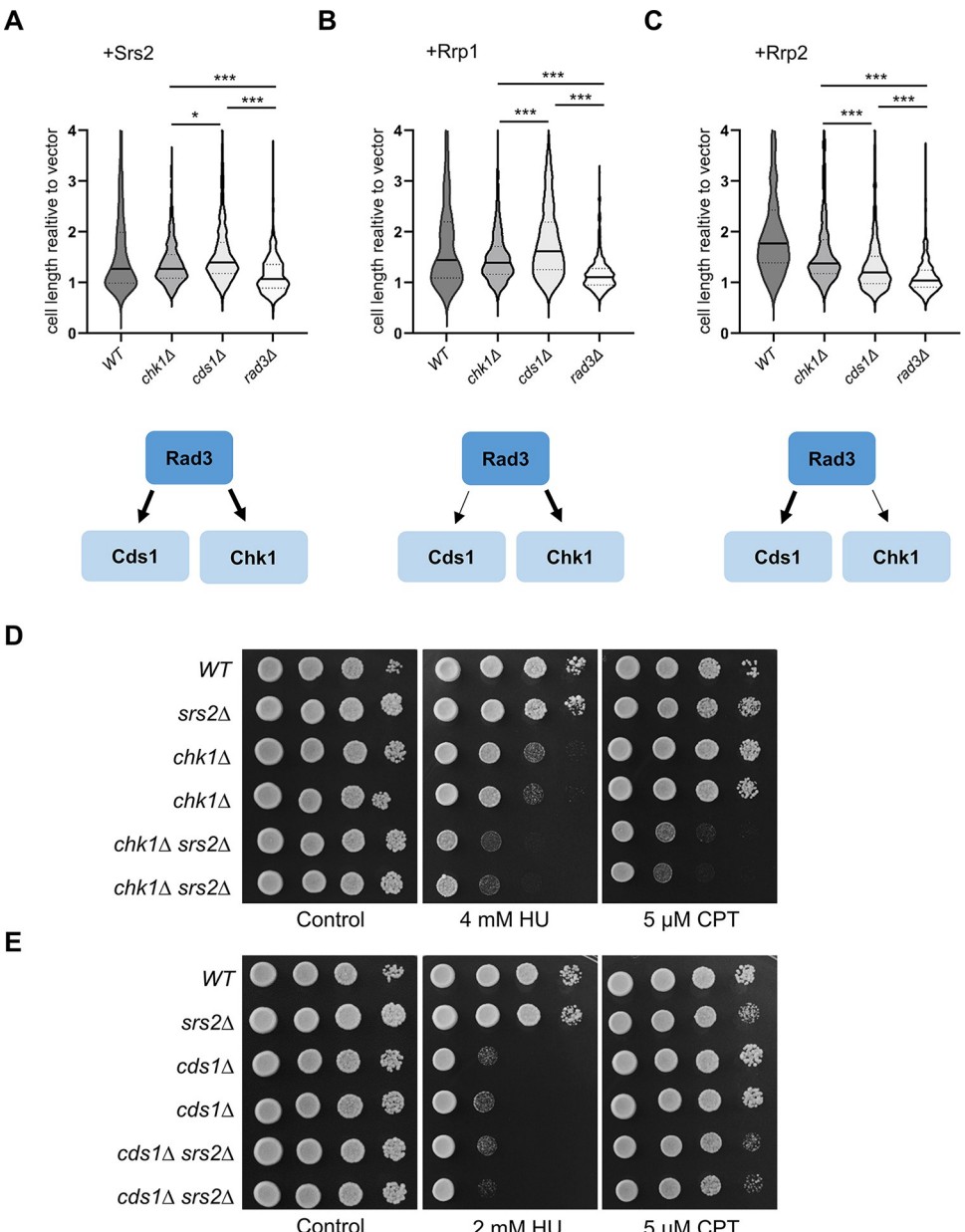

**Fig 6. DNA damage and replication stress checkpoint pathways are differently activated by the upregulation of Srs2, Rrp1 and Rrp2 protein levels.** Lack of Rad3, Chk1 and Cds1 differentially affects the length of cells over-expressing *srs2*+ (A), *rrp1*+ (B) and *rrp2*+ (C). Diagrams depicting major activated pathway are shown below. Transformants in respective deletion were grown for 48h in EMM under expression-inducing conditions (-leu), observed under the microscope and analysed with Axiovision rel. 4.8. Data from two independent transformations were analysed and the total number of cells counted was above 375. The centre line represents the median; the dotted lines represent upper and lower quartiles. Student's t-test was performed to calculate P-values (*** P $\leq$ 0.001, * 0.01 < P $\leq$ 0.05). Epistasis between srs2+ helicase and *chk1*+ (D) and *cds1*+ (E) genes. Cultures of respective single and *chk1Δsrs2Δ* and *cds1Δsrs2Δ* double mutant strains were appropriately diluted, spotted on YES plates with HU or CPT, incubated for 4 days and photographed.

[5]. Here, we show that this rescue also depends on the presence of Srs2 supporting our published model placing some aspects of Srs2 activity in a common pathway with the Rrp1-Rrp2 complex.

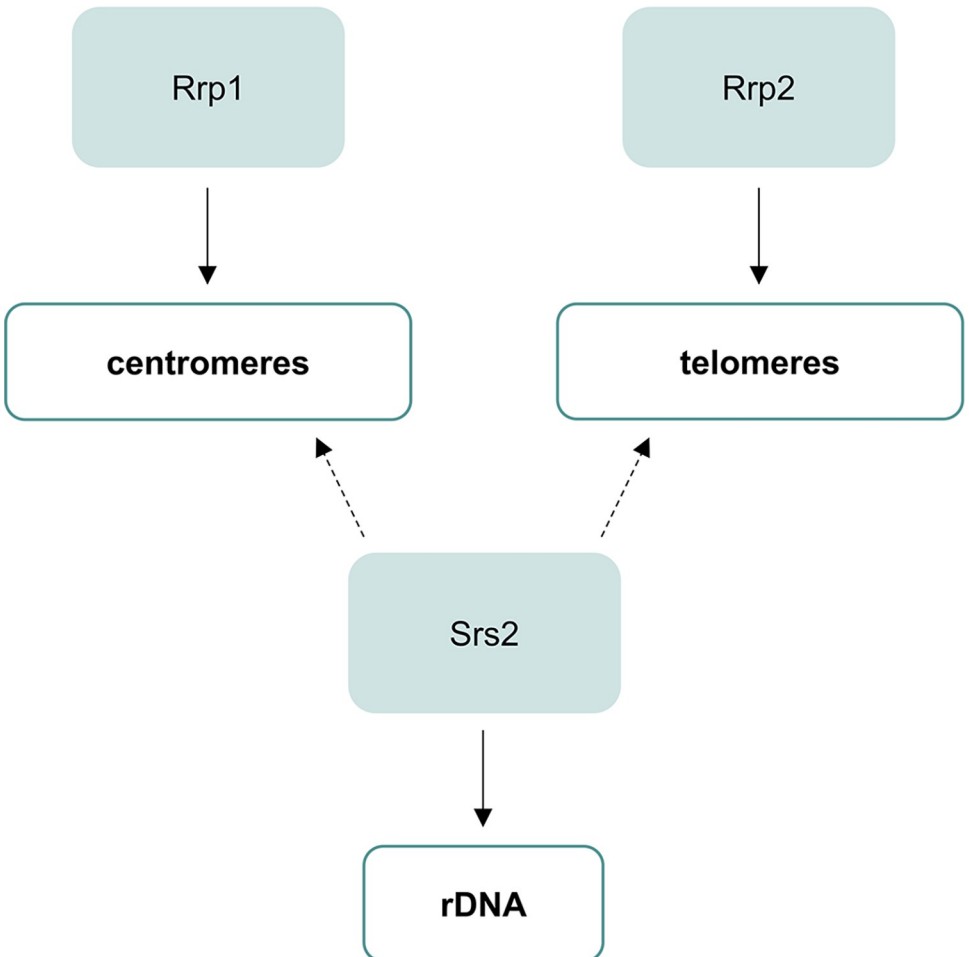

**Fig 7. A schematic model depicting involvement of Srs2 at rDNA and its possible impact on other genomic loci, like centromeres and telomeres, where Rrp1 and Rrp2 were shown to localize, respectively.**

Subsequently, by analysing the effect of upregulation of Rrp1 and Rrp2 protein levels we have found that they have activities independent from one another. Rrp1 is more important for centromere maintenance and its translocase and ubiquitin ligase activities are crucial for the regulation of Rad51 recombinase [7,9], while Rrp2 has been shown to protect cells from Top2-induced DNA damage and be required for telomere stability, a function not shared by Rrp1 [7,8]. Here we set out to analyse the effect of Srs2 overproduction.

We found that upregulation of Srs2 protein levels is associated with mild accumulation of ssDNA seen as an increase in the number of bright Rad11-EGFP foci, anaphase aberrations, chromosome instability and viability loss. These effects are similar to those seen for upregulation of Rrp1 levels but less severe than those caused by *rrp2+* overexpression. However, Srs2 localisation in the nucleus and checkpoint response to its overproduction appears to differ from Rrp1 and Rrp2.

We have previously demonstrated that overproduced Rrp1 and Rrp2 localise as distinct foci within the nucleus, and have been shown to be most abundant at centromeres and telomeres, respectively. Consistently, their activity was especially important during replication stress response at these respective regions [7]. Here, we demonstrate by fluorescence microscopy that Srs2 co-localizes with nucleolar chaperone Gar2 and confirm by chromatin

immunoprecipitation that Srs2 binds to the rDNA region. Moreover, our 2DGE analysis showed increased replication fork pausing at replication fork barriers within rDNA repeats in cells devoid of Srs2 helicase, highlighting its importance for unperturbed replication within this locus and hence for rDNA maintenance.

This is in line with previously published data showing Srs2 to be crucial for rDNA stability redundantly with the non-checkpoint function of Mrc1, a mediator of the replication checkpoint pathway [23]. Moreover, enhancing fork stalling at replication barriers has been demonstrated to be a replication checkpoint-independent function of Mrc1 [28]. We have shown that cells lacking Srs2 rely on functional DNA damage checkpoint for replication stress response as the double *chk1Δsrs2Δ* mutant was more sensitive to HU and CPT than the single *chk1Δ* mutant. One may thus hypothesize that Srs2 may participate in regulating the timing and speed of restart of replication forks stalled at repeats within the rDNA region, in order to prevent unscheduled events leading to DNA damage and the activation of Chk1-dependent checkpoint pathway. Upon dysregulation of Srs2 levels this process would be disturbed.

Our yeast-two-hybrid experiments suggest that Srs2 does not form a direct complex with either Rrp1 or Rrp2, but this does not preclude the existence of indirect interaction and/or collaboration between these proteins. We have observed that overproduced Srs2 forms distinct foci outside the nucleolus suggesting it has functions at other regions where blocks to replication may occur. Interestingly, Srs2 has been proposed to be required for a slow restart of DNA replication arrested at *RTS1*, a polar barrier present at *S. pombe* mating-type locus, but related to barriers found in the rDNA gene arrays as well as centromeres and telomeres [29]. One could then hypothesise that besides its role at rDNA, Srs2 may cooperate with Rrp1 and/or Rrp2 in response to replication stress at centromeres or telomeres (Fig 7).

It would be interesting in the future to determine to what extent the observed differences in checkpoint activation, as well as similarities and dissimilarities in other phenotypes induced by overproduction of Srs2, Rrp1 and Rrp2 described above, result from the differences in the place of action of these three proteins as well as their interacting partners. This would shed light on the specific requirements for the regulation of replication stress response pathways in different difficult-to-replicate regions in the *S. pombe* genome.

## Supporting information

**S1 File. S1 and S2 Figs and S1–S3 Tables.**
(PDF)

## Acknowledgments

We thank Matthew C. Whitby, Tony M. Carr, Jo Murray and Hiroshi Iwasaki for providing strains, Sarah Lambert for the protocol for examination of rDNA locus by two-dimensional gel electrophoresis and to Ireneusz Litwin for comments on the manuscript. The Gar2-mCherry strain was obtained from YGRC/NBRP Japan Resource Database (http://yeast.nig.ac.jp/yeast/top.xhtml).

## Author Contributions

**Conceptualization:** Dorota Dziadkowiec.

**Formal analysis:** Gabriela Baranowska, Dorota Misiorna, Wojciech Białek, Karol Kramarz, Dorota Dziadkowiec.

**Funding acquisition:** Gabriela Baranowska, Karol Kramarz.

**Investigation:** Gabriela Baranowska, Dorota Misiorna, Wojciech Białek, Karol Kramarz.

**Methodology:** Dorota Misiorna, Wojciech Białek, Karol Kramarz.

**Supervision:** Karol Kramarz, Dorota Dziadkowiec.

**Writing – original draft:** Karol Kramarz, Dorota Dziadkowiec.

**Writing – review & editing:** Gabriela Baranowska, Dorota Misiorna, Karol Kramarz, Dorota Dziadkowiec.

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
