## [Decision Letter · Decision Letter 0]

2 Jan 2024

PONE-D-23-39872Replication stress response in fission yeast differentially depends on maintaining proper levels of Srs2 helicase and Rrp1, Rrp2 DNA translocases.PLOS ONE

Dear Dr. Kramarz,

Thank you for submitting your manuscript to PLOS ONE. After careful consideration, we feel that it has merit but does not fully meet PLOS ONE’s publication criteria as it currently stands. Therefore, we invite you to submit a revised version of the manuscript that addresses the points raised during the review process. Please submit your revised manuscript by Feb 16 2024 11:59PM. If you will need more time than this to complete your revisions, please reply to this message or contact the journal office at plosone@plos.org. Please include the following items when submitting your revised manuscript:A rebuttal letter that responds to each point raised by the academic editor and reviewer(s). You should upload this letter as a separate file labeled 'Response to Reviewers'.A marked-up copy of your manuscript that highlights changes made to the original version. You should upload this as a separate file labeled 'Revised Manuscript with Track Changes'.An unmarked version of your revised paper without tracked changes. You should upload this as a separate file labeled 'Manuscript'.

We look forward to receiving your revised manuscript.

Kind regards,

Sudhir Kumar Rai, Ph.D

Academic Editor

PLOS ONE

Reviewers' comments:

Reviewer's Responses to Questions

**Comments to the Author**

1. Is the manuscript technically sound, and do the data support the conclusions?

Reviewer #1: Partly

Reviewer #2: Partly

Reviewer #3: Partly

2. Has the statistical analysis been performed appropriately and rigorously? 

Reviewer #1: Yes

Reviewer #2: Yes

Reviewer #3: Yes

3. Have the authors made all data underlying the findings in their manuscript fully available?

Reviewer #1: Yes

Reviewer #2: Yes

Reviewer #3: Yes

4. Is the manuscript presented in an intelligible fashion and written in standard English?

Reviewer #1: Yes

Reviewer #2: Yes

Reviewer #3: No

5. Review Comments to the Author

Reviewer #1: The study by Baranowska et al. genetically investigates the replication stress response in fission yeast and its dependence on maintaining proper levels of Srs2 helicase and Rrp1, Rrp2 DNA translocases. The upregulation of Srs2 helicase levels leads to enhanced replication stress, chromosome instability, and viability loss. Dysregulation of Srs2, Rrp1, and Rrp2 protein levels differentially affects checkpoint response. The study provides important insights into the regulation of replication stress and DNA repair processes in fission yeast.

Most data are of genetics, and sometimes it is hard to understand molecular relationship between factors, therefore authors should pay attention to illustrate molecular scheme underlying the genetic data, in text and in drawings.

Part of the conclusion has not been fully supported by the evidence provided in the submitted manuscript, which should be improved.

Specific comments are as listed below:

Introduction and Figure 1A,

Through introduction to Figure 1, It was not clear how Rqh1 is known to be involved in the Rad51-dependent HR events. Does Rqh1 work in the Rad55-Rad57 branch? In Figure 1A, rad57∆ is not sensitive to HU, but rqh1∆ is extremely sensitive, meaning that Rqh1 is involved in both Rad55 and SDSA pathways. Is this correct? It is required to explain the function of Rqh1 in the introduction and how we should interpret the spot assay showing intermediate sensitivity of rad57∆rqh1∆ against HU.

Figure 1B,

Evidence for the nuclei defects in photos (such as DAPI) is required.

Figures 2C and 3:

The labels should be “+thi” and “-thi” on the top, as expression levels change depending upon thiamine in the medium. This is a common issue throughout the figures presented in the manuscript.

Figure 2D,

Location of overexpressed EGFP-Srs2-FLAG needs to be clarified by use of a nucleolus marker at the same time with EGFP-Srs2.

p.7 l.204, “The double mutant of srs2 and rrp1 or rrp2 did not show any furhter delay,,, suggesting the role of Srs2 … may be independent of Rrp1 and Rrp2.”

I am a bit confused: the non-additive phenotype of srs2 rrp1/2 might indicate that those factors work in the same pathway rather than ‘independent’.

P.8 L..249; Figure 3C, D,

For me toxicity of overexpressed Srs2 was not evident in rrp1∆ and rad51∆ cells, but was still visible in rrp2∆ cells. Clear images are to be presented if author’s statement is correct.

Figure 3F,

1) When yeast two-hybrid results is ‘negative’, we cannot conclude that the two factors do not interact. This applies particularly when there are no positive/negative controls for the factors. In Figure 3F, no positive controls for Rrp1 or Srs2 have been used, therefore the results might simply reflect that the two-hybrid system per se in Figure 3F does not work with those proteins. Authors need to take the possibility into account in results or in discussion.

2) co-IP experiments using S. pombe cells are to be performed.

P.9 L.306 may affects > may affect

Figure 4C, D,

Images (such as DAPI) should be provided as evidence.

Figure 6,

As there are many factors appeared in the manuscript, and most data are of genetic analyses, a schematic summary may help understanding of readers.

Reviewer #2: The authors’ group previously reported that DNA translocases Rrp1 and Rrp2 act together with the Srs2 helicase in the common synthesis dependent strand annealing pathway of homologous recombination in fission yeast. In this manuscript, the authors further delved into their functions and revealed that these proteins are required for proper DNA replication completion after hydroxyurea (HU) treatment. As observed in cells overexpressing Rrp1 and Rrp2, the overexpression of Srs2 resulted in chromosome instability and enhanced replication stress. Moreover, the authors demonstrated that the overproduction of Rrp1 Rrp2, and Srs2 differentially activated DNA damage and replication stress checkpoint responses. Although many of the data presented in this manuscript are clear, some specific points need to be addressed prior to publication.

Specific points

1) Fig. 1A, to claim that Srs2 is required for the full rescue of the rqh1∆ mutant’s HU sensitivity by rad57∆, the authors should rule out a possibility that the growth defect of the srs2∆rqh1∆rad57∆ mutant is due to the HU sensitivity caused by the srs2∆ mutation.

2) In the epistasis analyses shown in Fig. 1C and D, no additive effect was observed in the completion of replication after HU treatment when the srs2∆ mutation is combined with either rrp1∆ or rrp2∆. These results strongly indicate that Srs2 functions in the same pathway with Rrp1 and Rrp2 rather than acting independently. Therefore, I think that the author’s conclusion “the role of Srs2 in resuming replication after HU block may be independent of Rrp1 and Rrp2 (Line 206)” does not reflect the authors’ findings properly.

3) In this manuscript, most of the experiments have been conducted using cells overexpressing Srs2, Rrp1 and Rrp2. Therefore, the author should confirm whether the expression level of those proteins is comparable to each other.

4) Fig. 6A-C, although the authors claim that cell length was decreased to a similar degree in the chk1Δ and cds1Δ mutants when Srs2 was overexpressed, it appears to me that at least the cell length of the cds1Δ mutant is increased. Moreover, the cell length of both chk1Δ and cds1Δ mutants overexpressing Rrp2 was significantly shortened, though the chk1Δ mutant was longer than cds1Δ. Therefore, the author’s statement in Lines 364-369 may need to be reconsidered.

Minor points

5) Fig. 2D, the nucleus should be detected by DAPI staining. Why the overexpression of Srs2 is so heterogeneous among cells?

6) Fig. 5C, the authors state that “DNA damage and replication checkpoint pathways may have redundant roles in augmenting the survival of cells overproducing Srs2 (Line321-324)”. This idea can be further confirmed by demonstrating that the Srs2 overexpression severely compromises cell growth of the chk1∆cds1∆ double mutant like in rad3∆.

Reviewer #3: In this manuscript Baranowska and coworkers assessed the role of DNA translocase Rrp1, Rrp2 and Srs2. These factors were previously described by this lab to regulate Rad51 recomibnase and to promote SDSA in response to replication stress and/or DNA damage (DSB). According to their model, Rrp1, Rrp2 and Srs2 function together and counteract Rad51 activity in the Swi5/Sfr1 sub-pathway of HR, favoring gene conversion rather than crossing over. In this new study, they investigated the impact of the overexpression of Srs2 with Rrp1 and Rrp2. High level of Srs2 activates simultaneously DNA damage and replication stress response and Srs2 localizes in the nucleolus.

The introduction does not clearly present current knowledge of the subject and does not properly lay the groundwork for this study. Plus, assumptions underlying this study have not been clearly established which makes this work hard to follow. Overall, the take home message of this study is not clear to me.

The genetic experiments are well conducted, however it would need further study to support their conclusions, regarding checkpoint activation as an example. They made the interesting observation that Srs2 accumulates in the nucleolus, suggesting a function of Srs2 in rDNA maintenance. Although it was previously established (Yasuhira , 2009), they did not further investigate the function of Srs2 in rDNA stability.

The lab previously proposed that Rrp1, Rrp2 and Srs2 function together, but they showed here that these factors may function distinctly and at different loci (telomeres, centromeres, rDNA). This point was not discussed. How can they reconciliate their first observations with their new results?

I have the impression that only the surface has been touched without a clear conclusion to this study. I am afraid that I can’t support the publication of this study at this stage.

Major points:

-The introduction does not clearly present current knowledge of the subject and does not properly lay the groundwork for this study. The authors should cautiously present the current understanding of HR regulation in S. pombe. Second paragraph of the introduction looks like a list of results without connection between them. Functions of of Rqh1 in the HR pathway should be properly introduced as well with its links with Rrp1/2. Distinction between observations and assumptions should be properly presented. The current status of the entire knowledge of Srs2 in S. pombe should be precisely established. A graphical abstract may help to clarify the entire introduction.

Figure 1:

Figure 1A: Survival curves should be presented as shown in their previous publication (NAR, 2013, Figure 6C). To fully support the epistatic function of Srs2 and Rrp1/2, the author should also test rad57� rqh1� srs2� rrp1� and rrp2� quadruple mutants with HU.

Figure 1C: For the WT, the authors claim that the doubling of intensity is due to the completion of replication, in this case number of cells should be double in the time course of the experiment (but not the mutants). Is that the case ? Accordingly, they should provide cell counts.

Figure 1D: “The double mutants simultaneously devoid of srs2+ and rrp1+ or

rrp2+ did not show any further delay in replication completion, suggesting the role of Srs2 in resuming replication after HU block may be independent of Rrp1 and Rrp2 (Fig. 1D).” I do not agree, it looks epistatic to me.

(minor point: the last paragraph lane 207-210 should be move to the next part lane 224)

Figure 2D:

EGFP-Srs2 localizes within the nucleolus. Does it affect rDNA stability? PFGE should be performed. 2D-gel analysis of the rDNA locus could have been assessed in srs2� mutant or when Srs2 is overexpressed to support their conclusions and also support their discussion (lane 425).

Figure 5:

Checkpoint activation is consistent with Rad11 foci accumulation. Chk1-phosphorylation is visible by western blot. As a control, treatment with DNA genptoxic agent should be shown to visualize phosphorylation of Chk1-HA and to evaluate the intensity of the phosphor shift. Why the phosphorylation status of Cds1 has not been monitored ? This should be done to support their claims (cf figure 6)

Figure 6:

“ Cell length was decreased to a similar degree in chk1Δ and cds1Δ mutants over-expressing srs2+ (Fig. 6A) suggesting that overproduction of Srs2 activated both DNA damage and replication stress response pathways.” According to the figure 6A, this is not the case. The effect of overexpression of Srs2 resembles to Rrp1, activating both checkpoints. I feel that these data do not support their conclusions. Failure in checkpoint activation in S-phase may lead to G2/M checkpoint activation.

6. PLOS authors have the option to publish the peer review history of their article (what does this mean?). If published, this will include your full peer review and any attached files.

Reviewer #1: No

Reviewer #2: No

Reviewer #3: No

---

## [Author Response · Author response to Decision Letter 0]

14 Feb 2024

RESPONSE TO REVIEWERS' COMMENTS 

General comments 

We would like to thank the Reviewers for their extensive and helpful comments. As suggested, we have performed additional experiments and rewritten parts of the text, and we hope the Reviewers will now find our manuscript satisfactorily improved.

Reviewer’s comments are in blue and our responses are in black font. Numbering of pages and figures refer to the revised version of the manuscript. All changes to the text of the manuscript are marked in yellow. 

Reviewer #1: The study by Baranowska et al. genetically investigates the replication stress response in fission yeast and its dependence on maintaining proper levels of Srs2 helicase and Rrp1, Rrp2 DNA translocases. The upregulation of Srs2 helicase levels leads to enhanced replication stress, chromosome instability, and viability loss. Dysregulation of Srs2, Rrp1, and Rrp2 protein levels differentially affects checkpoint response. The study provides important insights into the regulation of replication stress and DNA repair processes in fission yeast.

Most data are of genetics, and sometimes it is hard to understand molecular relationship between factors, therefore authors should pay attention to illustrate molecular scheme underlying the genetic data, in text and in drawings.

Part of the conclusion has not been fully supported by the evidence provided in the submitted manuscript, which should be improved.

Specific comments are as listed below:

Introduction and Figure 1A,

Through introduction to Figure 1, It was not clear how Rqh1 is known to be involved in the Rad51-dependent HR events. Does Rqh1 work in the Rad55-Rad57 branch? In Figure 1A, rad57∆ is not sensitive to HU, but rqh1∆ is extremely sensitive, meaning that Rqh1 is involved in both Rad55 and SDSA pathways. Is this correct? It is required to explain the function of Rqh1 in the introduction and how we should interpret the spot assay showing intermediate sensitivity of rad57∆rqh1∆ against HU.

We agree with the Reviewer that it was difficult to follow the data presented in the Figure 1 without description of the HR pathways in which all studied proteins are involved. We have now prepared a diagram of the fate of D-loop formed by Rad51 with the help of two mediator complexes and marked positions where Rqh1, Srs2 and Rrp1/2 are thought to be involved. We placed it Supplementary Figure S1 together with the data introducing epistasis analysis between rad57∆rqh1∆ with rqh1∆ that refer directly to our previous analysis with rrp1∆ and rrp2∆ [1]. We hope that currently the figure is clear and constitutes a comprehensive starting point for data presented in this manuscript.

Figure 1B,

Evidence for the nuclei defects in photos (such as DAPI) is required.

We have now added, as Figure S1D, the representative photographs of DAPI stained cultures of 3 discussed mutants and indicated the examples of lagging, cut and mis-segregated chromosomes in the photograph depicting rqh1Δ mutant cells. 

Figures 2C and 3:

The labels should be “+thi” and “-thi” on the top, as expression levels change depending upon thiamine in the medium. This is a common issue throughout the figures presented in the manuscript.

We changed the labelling of Figures throughout the manuscript, as requested. 

Figure 2D,

Location of overexpressed EGFP-Srs2-FLAG needs to be clarified by use of a nucleolus marker at the same time with EGFP-Srs2.

We re-examined localisation of overexpressed EGFP-Srs2-FLAG in a strain expressing Gar2-mCherry, a nucleolar histone chaperone, and included these data in new Fig.3A. As before, we observed diffused localisation of Srs2 within nucleolus and also foci, often located outside nucleolus. We thus extended description of obtained data (lines 344-353). In order to confirm microscopy data, we performed ChIP experiment, now included as Fig. 3B, with description, line 351-353:

 “Finally, we have confirmed by chromatin immunoprecipitation of EGFP-Srs2 followed by qPCR analysis that when overproduced Srs2 accumulates at DNA within rDNA region”.

p.7 l.204, “The double mutant of srs2 and rrp1 or rrp2 did not show any furhter delay,,, suggesting the role of Srs2 … may be independent of Rrp1 and Rrp2.”

I am a bit confused: the non-additive phenotype of srs2 rrp1/2 might indicate that those factors work in the same pathway rather than ‘independent’.

We are very grateful that this and other Reviewers pointed out this obvious mistake we made. We wrote precisely the opposite of what we meant to write. The whole point of overexpression analysis performed later was to examine if Srs2 has roles independent from Rrp1 and/or Rrp2. The relevant sentence now reads, line 255-258: 

“The double mutants simultaneously devoid of srs2+ and rrp1+ or rrp2+ did not show any further delay in the entry of chromosomes into the gel (Fig. 1B) suggesting that Srs2, Rrp1 and Rrp2 may have overlapping roles in the resolution of replication intermediates after HU block.”

P.8 L..249; Figure 3C, D,

For me toxicity of overexpressed Srs2 was not evident in rrp1∆ and rad51∆ cells, but was still visible in rrp2∆ cells. Clear images are to be presented if author’s statement is correct.

We substituted photographs of same plates after longer incubation, and we hope now the difference between the growth of srs2+ expressing cells and empty vector controls are more evident. We wish to point out that growth defect observed upon Srs2 overproduction, as well as Rrp1, is not very pronounced, and is often more readily observed on plates rather as smaller colonies than marked decrease in their number. We hope that when the Reviewer compares the effect of Srs2 overproduction in wild-type (Fig.2A) to that in respective mutants (Fig. 2D, E), they will agree that toxicity of Srs2 is the same in wild-type and rrp1Δ mutant and only slightly more pronounced in cells of rrp2Δ and rad51Δ mutants.

Figure 3F,

1) When yeast two-hybrid results is ‘negative’, we cannot conclude that the two factors do not interact. This applies particularly when there are no positive/negative controls for the factors. In Figure 3F, no positive controls for Rrp1 or Srs2 have been used, therefore the results might simply reflect that the two-hybrid system per se in Figure 3F does not work with those proteins. Authors need to take the possibility into account in results or in discussion.

2) co-IP experiments using S. pombe cells are to be performed.

We agree with the Reviewer that negative Y2H results do not prove lack of interaction between proteins, but we only argued in the text that direct interaction was probably absent. 

We do not show controls with constructs used, but Y2H interaction between Rrp1 and Rrp2 as well as SUMO, was shown by us before using same constructs [2]. We have data on possible Srs2 self-interaction, pasted below, showing that this construct is also functional. We do not wish, however, to discuss this result in present manuscript. (Please note that for Rrp1 and Rrp2 on pGAD the growth on -3 medium results from self-activation due to the ability of Rrp1 and Rrp2 to bind to DNA.)

We performed co-IP experiments as suggested by the Reviewer. We obtained very preliminary result suggesting that EGFP-Srs2 may co-immunoprecipitate with Rrp1-HA (see red square in Western blot pasted below). Unfortunately, the signal was very weak and no reciprocal IP (Rrp1-HA with anti-GFP antibody) was observed. The result with Rrp2 was nonconclusive. 

These are clearly technical issues and extensive optimisation of experimental conditions is required which we will certainly undertake. However, this will take time and would delay the publication of our findings. Therefore, we decided to emphasised this point in our description of these data, line 324-327: 

“It is important to note, however, that such interaction may be mediated through other proteins, and needs to be investigated further. Nevertheless, the above data indicate that the effects of overexpression of these three genes may be, at least in part, independent of each other”. 

We hope that the Reviewer agrees that for the purpose of this manuscript this experiment is not indespensible.

P.9 L.306 may affects > may affect

We corrected this mistake.

Figure 4C, D,

Images (such as DAPI) should be provided as evidence.

We included a representative photograph of wild-type cells overproducing Srs2-EGFP in Fig. 4C. 

For chromosome assay, however, this is not practical. As stated in Materials and Methods, we counted the number of red among many white colonies on numerous YES low Ade plates, these events are not frequent. We do not have photographs of all those plates and hope that adding to figure a schematic diagram of the assay, together with the explanation, line 392-393: 

”Cells that lose this mini-chromosome accumulate red intermediate due to the inability to complete adenine synthesis (Fig. 4E).” 

will satisfy the Reviewer.

Figure 6,

As there are many factors appeared in the manuscript, and most data are of genetic analyses, a schematic summary may help understanding of readers.

We included a diagram of discussed HR pathways as Fig. 1SA and schematic summary of our conclusions as Fig. 7.

Reviewer #2: The authors’ group previously reported that DNA translocases Rrp1 and Rrp2 act together with the Srs2 helicase in the common synthesis dependent strand annealing pathway of homologous recombination in fission yeast. In this manuscript, the authors further delved into their functions and revealed that these proteins are required for proper DNA replication completion after hydroxyurea (HU) treatment. As observed in cells overexpressing Rrp1 and Rrp2, the overexpression of Srs2 resulted in chromosome instability and enhanced replication stress. Moreover, the authors demonstrated that the overproduction of Rrp1 Rrp2, and Srs2 differentially activated DNA damage and replication stress checkpoint responses. Although many of the data presented in this manuscript are clear, some specific points need to be addressed prior to publication.

Specific points

1) Fig. 1A, to claim that Srs2 is required for the full rescue of the rqh1∆ mutant’s HU sensitivity by rad57∆, the authors should rule out a possibility that the growth defect of the srs2∆rqh1∆rad57∆ mutant is due to the HU sensitivity caused by the srs2∆ mutation.

The way we understand the epistatic analysis between these three genes, both these statements are not exclusive, but are different ways of describing the same phenomenon.

Numerous studies have demonstrated that srs2∆ shows strong synthetic growth defect/is synthetic lethal with rqh1∆ mutant, demonstrating that these helicases act on alternative, at least partially independent pathways, and at least one is required for viability. This is suppressed by deleting rhp55+ or rhp57+ [3], showing that it is the accumulation of Rad57-dependent recombination intermediates that is toxic in the double rqh1∆srs2∆ mutant. 

Here we focus on HU as one specific, source of DNA replication induced recombination intermediates. We cannot asses the sensitivity to HU of rqh1∆srs2∆ double mutant due to its huge growth defect, but by examining the sensitivity of the rqh1∆srs2∆rad57∆ mutant, we can uncover that both deletion of srs2+ and rqh1+ contribute. Thus, HU sensitivity of rqh1∆srs2∆rad57∆ mutant, which is greater than sensitivity of rqh1∆rad57∆ mutant, results from HU sensitivity of both rqh1∆ and srs2∆. This means that both helicases are involved in dealing with HR intermediates generated by HU treatment. In other words, Srs2 mediated alternative pathway for dealing with HU induced recombination intermediates allows the full rescue of rqh1∆ HU sensitivity by deletion of rad57+.

2) In the epistasis analyses shown in Fig. 1C and D, no additive effect was observed in the completion of replication after HU treatment when the srs2∆ mutation is combined with either rrp1∆ or rrp2∆. These results strongly indicate that Srs2 functions in the same pathway with Rrp1 and Rrp2 rather than acting independently. Therefore, I think that the author’s conclusion “the role of Srs2 in resuming replication after HU block may be independent of Rrp1 and Rrp2 (Line 206)” does not reflect the authors’ findings properly.

As we wrote in response to the Reviewer #1 we are very grateful for pointing out this obvious mistake we made. We wrote precisely the opposite of what we meant to write. We changed the wording and the relevant sentence now reads, line 255-258: 

“The double mutants simultaneously devoid of srs2+ and rrp1+ or rrp2+ did not show any further delay in the entry of chromosomes into the gel (Fig. 1B) suggesting that Srs2, Rrp1 and Rrp2 may have overlapping roles in the resolution of replication intermediates after HU block.”

3) In this manuscript, most of the experiments have been conducted using cells overexpressing Srs2, Rrp1 and Rrp2. Therefore, the author should confirm whether the expression level of those proteins is comparable to each other.

We now included in Fig. 2C the Western blot comparing Srs2, Rrp1 and Rrp2 protein levels in two different transformants. Srs2 seems more abundant, but the protein levels do not corelate directly with growth defect as overproduction of Rrp2 is more toxic than Rrp1 while the levels of both proteins are comparable. 

4) Fig. 6A-C, although the authors claim that cell length was decreased to a similar degree in the chk1Δ and cds1Δ mutants when Srs2 was overexpressed, it appears to me that at least the cell length of the cds1Δ mutant is increased. Moreover, the cell length of both chk1Δ and cds1Δ mutants overexpressing Rrp2 was significantly shortened, though the chk1Δ mutant was longer than cds1Δ. Therefore, the author’s statement in Lines 364-369 may need to be reconsidered.

This point was also raised by Reviewer #3 so we realise we did not express ourselves clearly and seemed to imply that only one checkpoint pathway was activated by overproduction of studied proteins. We have now rewritten description of data presented in Fig. 6A-C, (lines 477-490), and redesigned complementing diagrams to emphasise that we identify the differences in the prevailing pathway activated in Srs2, Rrp1 and Rrp2 overproducing cells. 

Minor points

5) Fig. 2D, the nucleus should be detected by DAPI staining. Why the overexpression of Srs2 is so heterogeneous among cells?

In S. pombe DAPI staining gives best results on fixed cells but compromises the quality of GFP signal.

Instead we examined localisation of overexpressed EGFP-Srs2-FLAG in a strain expressing Gar2-mCherry, a nucleolar histone chaperone, and included these data in new Fig. 3A. We confirmed microscopy data by ChIP analysis, presented in Fig. 3B.

The cultures examined are not synchronised, so cells are at different stages of cell cycle. Additionally, the expression system we employ does not allow to precisely regulate protein levels (differences in plasmid copy number in the cell, plasmid loss, long induction time for nmt promoter, etc), and for that reason we always examine at least several independent transformants.

6) Fig. 5C, the authors state that “DNA damage and replication checkpoint pathways may have redundant roles in augmenting the survival of cells overproducing Srs2 (Line321-324)”. This idea can be further confirmed by demonstrating that the Srs2 overexpression severely compromises cell growth of the chk1∆cds1∆ double mutant like in rad3∆.

Rad3 is an upstream kinase to both, Chk1 and Cds1, so we concluded that analysing rad3Δ mutant would suffice. We agree with the Reviewer that several experiments could be done to corroborate our data and it would be interesting to explore in greater detail checkpoint dependent and independent interactions of Srs2 with Chk1, Cds1 and Rad3 but we hope the Reviewer agrees that this falls outside the scope of this work.

Reviewer #3: In this manuscript Baranowska and coworkers assessed the role of DNA translocase Rrp1, Rrp2 and Srs2. These factors were previously described by this lab to regulate Rad51 recombinase and to promote SDSA in response to replication stress and/or DNA damage (DSB). According to their model, Rrp1, Rrp2 and Srs2 function together and counteract Rad51 activity in the Swi5/Sfr1 sub-pathway of HR, favoring gene conversion rather than crossing over. In th

---

## [Decision Letter · Decision Letter 1]

28 Feb 2024

Replication stress response in fission yeast differentially depends on maintaining proper levels of Srs2 helicase and Rrp1, Rrp2 DNA translocases.

PONE-D-23-39872R1

Dear Dr.Karol Kramarz,

We’re pleased to inform you that your manuscript has been judged scientifically suitable for publication and will be formally accepted for publication once it meets all outstanding technical requirements.

Kind regards,

Sudhir Kumar Rai, Ph.D

Academic Editor

PLOS ONE

Reviewers' comments:

Reviewer's Responses to Questions

**Comments to the Author**

1. If the authors have adequately addressed your comments raised in a previous round of review and you feel that this manuscript is now acceptable for publication, you may indicate that here to bypass the “Comments to the Author” section, enter your conflict of interest statement in the “Confidential to Editor” section, and submit your "Accept" recommendation.

Reviewer #2: All comments have been addressed

2. Is the manuscript technically sound, and do the data support the conclusions?

Reviewer #2: Yes

3. Has the statistical analysis been performed appropriately and rigorously? 

Reviewer #2: Yes

4. Have the authors made all data underlying the findings in their manuscript fully available?

Reviewer #2: Yes

5. Is the manuscript presented in an intelligible fashion and written in standard English?

Reviewer #2: Yes

6. Review Comments to the Author

Reviewer #2: (No Response)

7. PLOS authors have the option to publish the peer review history of their article (what does this mean?). If published, this will include your full peer review and any attached files.

Reviewer #2: No
